# Using Technology the Right Way to Support Social Connectedness for Older People in the Era of COVID-19

**DOI:** 10.3390/ijerph18168725

**Published:** 2021-08-18

**Authors:** Louise McCabe, Alison Dawson, Elaine Douglas, Nessa Barry

**Affiliations:** 1Faculty of Social Sciences, University of Stirling, Stirling FK9 4LA, UK; a.s.f.dawson@stir.ac.uk (A.D.); elaine.douglas@stir.ac.uk (E.D.); 2Technology Enabled Care and Digital Healthcare Innovation, Scottish Government, Edinburgh EH12 9EB, UK

**Keywords:** older people, technology, human rights, social connectedness

## Abstract

The restrictions imposed in response to the COVID-19 pandemic pose significant risks to the human rights of older people from limitations in how people are able to engage with their social lives and from increased risk of discrimination linked to conceptualization of COVID-19 as a disease of the old. Further, COVID-19 increases risks of social isolation through public health and societal responses such as lockdowns. These responses have resulted in significant shifts in how citizens and service providers think about technology as a tool to allow people to stay socially connected. However, there are risks to the rights of older people inherent in the use of technology related to their ability to access technology and ageist assumptions that may limit engagement. The ‘Technology and Social Connectedness’ project was a pre-pandemic mixed-methods study involving evidence review, secondary analyses, and qualitative methods. Cross-dataset analyses led to evidence-based guidance to inform a rights-based approach to using technology. This paper provides analysis from the project that foregrounds a rights-based approach demonstrating how we developed the guidance within this framework and, contextualized within the pandemic response in Scotland, how that guidance can help others to protect and uphold the human rights of older people.

## 1. Introduction

Human rights, the basic rights and freedoms to which all people are entitled, include rights to equal access to services and to equal participation in economic, social, cultural, and leisure activities. However, COVID-19 directly and indirectly impinges on older people’s enjoyment of these rights [1]. The likelihood of experiencing greater severity of COVID-19 symptoms and the risk of death from COVID-19 increases with age [2]. The incidence of a range of health conditions which would make individuals more vulnerable to COVID-19 increases with age, resulting in higher numbers of older people being advised, or feeling compelled, to remain in their homes during the pandemic. This presents increased risks of social isolation and loneliness, particularly in older people who tend to have smaller social networks [3]. Reductions in the frequency of social contacts resulting from government-mandated or self-imposed restrictions on social activities may result in increased loneliness, with concomitant negative impacts on health and wellbeing. Research conducted as the pandemic took hold suggests that in many countries loneliness in older people increased, e.g., in Scotland [4], the UK [5], the Netherlands [6], Italy [7], Japan [8], and that increased loneliness was associated with worsening wellbeing and health [4].

Services and interventions enabled by digital technologies may help people to maintain or even increase pre-pandemic levels of social connectedness. There are gaps and issues with the evidence base around the efficacy of technology-enabled interventions in maintaining social connection and reducing loneliness which McCabe et al. [9] identified in the study which informs this paper and which are summarised below, but there is very recent evidence that prior knowledge of using social networking sites (SNSs) and access to them during lockdown helped some older people to feel more socially included [7] and that access to SNS and other digital services during lockdown was valued [5].

Commentators [10,11] have highlighted the potential role for technology-based interventions to support social connection whilst physical distancing is required to limit the spread of COVID-19 and have called for solutions to enable older people to make use of existing services. Digital poverty, digital literacy, hardware compatibility, and user accessibility are just some of the challenges which need to be addressed if new services and interventions are to avoid compounding COVID-driven inequalities for older people.

Taking a rights-based approach to the design and delivery of technological interventions will help to ensure that technology-enabled services or interventions designed to ameliorate social isolation will also act to redress rather than exacerbate the current COVID-related impingement of older people’s rights to participate fully in all aspects of social and civic life. The T&SCon guidance is designed to support a rights-based approach to delivering such interventions and the following sections describe the analysis underpinning the development of the guidance, undertaken before the pandemic, and reflect on its relevance within the current context of continuing restrictions and fears about COVID-19.

### Context for Guideline Development

Even before the COVID-19 pandemic, social isolation was recognised as a growing challenge in Scotland and elsewhere. In the 20 years leading up to 2019 the number of people living alone in the UK increased by about 20 percent from 6.8 to 8.2 million people [12] and a similar pattern is seen in many countries around the world [13]. In 2017 in Scotland, around 885,000 people were living alone, of whom 350,000 were aged 65 years or older [14].

Social isolation has been found to impact negatively on the wellbeing of older people, increasing risks of a range of physical and psychological health conditions (see, e.g., [15,16]). The need to increase social connection in order to reduce these risks is recognised in policy across the UK, with England and Scotland separately setting out aims and objectives to address social isolation, including through the use of technology [17,18]. Recent responses to the COVID-19 pandemic across many countries have led to the rapid adoption of technology to support social connectedness in many new ways and demonstrate interest in the potential for technology to play a positive role.

Research suggests a positive link between the use of technology and social connectedness. Studies using cross-sectional surveys or secondary analysis of survey data demonstrate a positive correlation between current use of ICT and social connectedness [19,20,21,22]. Other studies examining the use of social networking sites such as Twitter also show positive correlations between the use of these sites and social connectedness [23,24,25,26,27]. Overall, such studies tend to suggest a positive relationship between ICT/internet use and social connectedness, but data collected at a single time point do not provide any insight into causality. It is not clear whether people who use ICT and internet-based communications become more socially connected *as a result of* that use or whether people who were already connected are more likely to use technology.

Moreover, many of these studies suggest ‘digital divides’, e.g., in relation to age, employment status/occupation, household income and education). They also reveal possible age-related differences in effects and underscore the importance of perceived usefulness of online social networks, perceived knowledge and self-efficacy in ICT use, and the accessibility and usability of hardware and software interfaces in influencing social network site use in different populations. These findings are mirrored in more recent research undertaken during the pandemic as outlined above (e.g., [5,7]). Due to speed at which COVID19 hit, many initiatives were implemented in haste and less is known about how activities which became widespread during the pandemic uphold and protect the human rights of older people within rapidly changing patterns of lockdowns and restrictions.

It is globally acknowledged that the human rights of older people are often ignored and violated [28]. In Scotland there is growing consensus that care and support for older people should be delivered within a rights-based framework [29,30,31]. A rights-based approach underpins current Scottish policy for health and social care, most notably in the *Health and Social Care Standards* [32] that promote care that supports dignity, respect, compassion, and participation.

This paper focuses on human rights set out in Articles 8 and 14 of the Human Rights Act (1998), UK legislation which gives effect to effect to rights and freedoms guaranteed under the European Convention on Human Rights (See Figure 1 below). Article 8 sets out the right to a private life and ensures respect for private and family life. This includes the right for everyone to participate in economic, social, cultural, and leisure activities and recognises that sometimes people might need help to do that. When thinking about technology, that help might for example be in the form of universally available internet access which, during the pandemic, has proved critical to supporting many peoples’ access to family life and participation in cultural and leisure activities. Article 8 emphasises the right to ‘a *private* life’; it is crucial that where it is employed to sustain and increase social connection, technology is used in a way that holds information about individuals securely and in line with legal requirements to ensure privacy and reduce risk and surveillance.

Article 14 reminds us about the right to protection from discrimination, which occurs when some people are treated less favourably than others in a similar situation. Older people face ageism in many forms and may face negative stereotyping in regard to their engagement with and use of technology.

A rights-based approach to using technology will ensure that older people are treated fairly, that risk for them is reduced and that they can use technology in a meaningful way to access family life and participate in cultural and leisure activities. This continues to be crucial, as pandemic associated restrictions on social life look to continue for some time as we adjust in a peri- and post-COVID19 world.

The T&SCon project, reported here, took a rights-based approach to explore the potential of technology to address and ameliorate social isolation for older adults living in Scotland in 2019. The authors investigated the range of factors that influence technology use by older people to facilitate engagement with new forms of social connection. The project focused specifically on social connectedness to shed light on positive outcomes and how they can be achieved.

The overarching aim of this paper is to inform discussion on how technology can be used in ways that protect and uphold the human rights of older people. Its specific objectives are: (1) to demonstrate how we used findings from the T&SCon project to build co-created guidance that supports individuals and organisations to take a rights-based approach to using technology to support social connectedness; (2) to present the project findings using the PANEL framework [33], to demonstrate where current practice aligns with a rights-based approach and where there are gaps or areas for development; and (3) to illustrate how insights from use of the PANEL framework informed the final structure and content of the guidance and how as a result it can be used to prompt consideration of risks to older people’s human rights during the pandemic and beyond.

The PANEL (Participation, Accountability, Non-Discrimination and Equality, Empowerment and Legality) principles are an internationally recognised framework to understand the application of a human rights-based approach in practice, including in Scotland, where they have been applied by the Scottish Human Rights Commission in a variety of health and social care contexts [33,34]. The PANEL approach is used here to assess alignments with a rights-based approach in practice as well as identify gaps or areas for development for using technology to support social connectedness.

The Scottish health and care system, which promotes working in partnership between health, local authority social care, independent and third sector partners, provides the setting for discussion, prior to COVID-19 and amplified during the COVID-19, about the application of a rights-based approach in health and care.

## 2. Materials and Methods

Data presented in this paper are drawn from the T&SCon Project. The T&SCon project was a mixed methods study undertaken in 2019 [35,36] with four specific components: a review of evidence; secondary analysis of the Health Ageing in Scotland (HAGIS) dataset [37]; focus groups with a range of stakeholders; and co-creation workshops to refine the findings from these three activities (see Figure 2).

### 2.1. Review of Evidence

The review of evidence had two components: a systematic literature review and an internet scoping of technology project. The literature review was conducted to assess and report on the research evidence base for the effectiveness of technology enabled interventions to improve social connectedness for adults. The review was complemented by a technology scoping exercise that involved a wide-ranging internet search for current projects and initiatives using technology to support social connectedness.

### 2.2. Focus Groups

A series of four focus groups were held with 25 stakeholders from across Scotland including six health and social care practitioners, eight members of staff from third sector organisations providing care and housing support, one member of staff from a technology company and ten older people. Discussion at the focus groups was about the experiences of participants in using technology to promote social connectedness; the challenges experienced in delivering projects and in using technology; and reflections on the current and potential use of technology to support social connectedness. Focus groups were audio recorded and fully transcribed. All participants provided written informed consent. The older participants were aged between 60 and 80: three lived in urban areas and seven lived in rural areas of Scotland.

### 2.3. Secondary Analysis of Healthy Ageing in Scotland (HAGIS)

HAGIS data were gathered pre-pandemic (2016-17) as part of the HAGIS pilot study [37]. Secondary analyses were conducted to understand the relationship between social connectedness, loneliness, and use of technology in an older population in Scotland.

### 2.4. Co-Production Workshops

Two co-production workshops were conducted involving 13 participants including 10 health and social care practitioners together with a person with dementia, a person with mental health issues living in supported housing and an older person with sight loss. The workshops involved a series of activities that supported reflection on and refinement of findings from earlier parts of the project. Fieldnotes and written contributions from participants were collected during the workshops. All participants provided written informed consent.

A full description of the methods and participant details are provided in the project report [9]. Quotations provided in the findings below include a code that indicates if it is taken from a focus group (e.g., FG01) or co-production workshop (e.g., WS01) and a participant number indicating whether it is a professional participant (e.g., P01) or an older person (e.g., OP01).

### 2.5. Approach to Data Analysis

The secondary data analysis carried out in the T&SCon project sought to understand the relationship between social connectedness, loneliness, and use of technology in a representative sample of older people living in Scotland using data from the Healthy Ageing In Scotland study [37]. Social connectedness was measured by asking respondents how often, and by which means they communicate with the following social groups: children, family, and friends. Technology use was measured using a series of questions on the frequency and function of internet use, where people accessed the internet, and which type of device they used. We then tested for association between the profiles of social connectedness and (i) population characteristics (ii) loneliness, and (iii) internet use. See the full project report for more detail [9].

PANEL principles formed the framework of our analyses to enable exploration of how and to what extent technology is used in a manner that supports the rights of older people (see Figure 3).

## 3. Results

During periods of pandemic-related restriction, the right to participate in a family life is curtailed for many older people and technology is providing one way to support this participation. Data from our project, collected before the pandemic, confirm that older people are using technology in diverse ways to stay socially connected. However, it is important to understand that older people are individuals and will make choices about who they stay in touch with and how frequently.

Secondary data analyses derived six distinct profiles of social connectedness in the older Scottish population. Those who were highly connected with all social groups (High: All); particularly high with children (High: Children); and Friends (High: Friends) are represented in blue (see Figure 4, left) and those with No Children, No Other Family, and No Friends and 1/3 No Children are represented in green. The size of the profile in Figure 4 is proportionate to the underlying sample size. That is, 40% of the sample are in the High: All profile and 6% of the sample are in the No Friends profile.

Key distinctions between patterns of social isolation and loneliness in older people were found within our sample. This is visually represented where the profiles of social connectedness (left, Figure 4) are associated with self-reported loneliness (right, Figure 4). As shown, some of those with the highest level of social connectedness also experienced loneliness, while some of those with fewer and less frequent social connections said they never experienced feelings of loneliness. This distinction between the objectivity of social connections and the subjectivity of loneliness is very important when considering human rights to participation. Upholding rights to participation means giving older people choice with regard to the frequency, purpose and mode of interactions. There is no prescribed or set level of social connectedness.

We found many of the older people in our sample to use the internet or email to some extent and to predominantly access it from their own homes. Those who hardly ever or never use the internet were more likely to be older than those who use it often. The most common use of the internet in older people is to send or receive email, find information on goods or services, and shopping. These activities varied by social connection profile. Those with the fewest social connections were also least likely to use the internet for email or to seek information which is likely to have had implications for how these people could use technology to stay connected during pandemic restrictions. It was also noted that those with regular contact with their children (High: Children) were less likely to use the internet for these activities. It may be that their children could have used the internet on their behalf for information seeking or shopping prior to and/or during pandemic restrictions. It is also possible, that with such close connections to their children, that they may have been supported to adopt the internet for these purposes during pandemic restrictions. It is feasible that the impact of the pandemic on these two profiles could have diverged given the difference in levels of social support available to them pre-pandemic.

Primary data from the T&SCon project demonstrated that technology was a useful mechanism pre-pandemic to support participation within established social networks and to connect with new people in a social way.

Older people reported using online services to connect with families across generations and around the world. The low cost of connecting using online methods allowed families to have longer periods together and to adopt more informal types of interaction, for example, continuing with cooking and mealtimes while connected to others online.


*I have twin sons, one lives in Japan and the one lives in Hong Kong…and every Sunday afternoon we switch on messenger or FaceTime and the three of us, I don’t stop and sit to talk to them, just get on with what I am doing (FG1OP02)*


Video calling appeared to offer a closer connection between family members than a simple phone call. One older person reported that her daughter no longer felt the need to pop in everyday if she could ‘see’ her mum online and be reassured that she was okay. This was seen as positive by the older person, who retained more independence.

For others, being able to see their relative was important in terms of assessing their wellbeing. One example given was of a daughter who noticed her father stopped turning the video on during calls. This prompted her to visit in person and discover that he was badly bruised following a fall.

Older people also used social media sites to stay connected to families and to wider communities.


*I use it [Facebook] for quite a lot, mainly actually because five of my grandkids don’t live in [place]…I was using it to help them with their homework…plus I’ve got a lot of relatives in the States so I was actually keeping in contact with them (FG4OP04)*


Social media sites were also accessed to find people locally with similar interests and to get involved in activities, such as a women’s walking group or connecting with people with similar interests in other parts of the world. It was interesting to see that when these online activities were ranked by workshop participants it was felt by them that connecting with people and finding opportunities locally was more important than accessing the wider world.

Technology was also used to support connection between people with similar health conditions, sometimes supported by health and social care staff, and utilising health-service based technology that was perceived by users and providers as more secure when sharing health-related information.

*We had a gentleman living in [island area] who wanted to set up a stroke group but he wasn’t able to physically get out of the house to do that, he already had a laptop there…so I was able to go down and get him set up with Attend Anywhere for a monthly group (FG02P01)* (‘Attend Anywhere’ is the proprietary name of the web-based video consultation platform which is currently used in Scotland. Since 2019 the service has more commonly been referred to as ‘NHS Near Me’ or ‘Near Me’).

Technology also worked indirectly to support wellbeing of older people and enable their participation in social life. Numerous examples were given of technology being used in place of or to complement in-person medical appointments. The use of technology was associated with reduced travel times but more importantly in reduction of stress associated with travelling, waiting for, and attending appointments.

Our data show that pre-pandemic older people were using technology in a variety of ways to stay connected to their family and cultural lives.

This section now goes on to present findings from the analysis of primary data collection from the focus groups and co-creation workshops and utilises the PANEL principles to frame the analysis. The findings illustrate how experiences of older people and professionals in Scotland align with a rights-based approach to the use of technology and where there are lessons to learn. In the discussion section that follows we draw these findings and those from the evidence review and secondary analysis together to demonstrate the development of the T&SCon guidance.

### 3.1. Participation

The P in the PANEL principles asks us to examine how people are included in decision-making that may impact on their rights, leading us to consider how older people are supported to participate in decisions directly relating to the design of technology or the manner in which it is used.

The T&SCon data suggest that the extent to which organisations were (or were not) including older people and other key stakeholders such as frontline staff in using technology impacted on successful implementation. There was broad agreement among participants that technology companies are not doing enough to engage with potential end users and support them to participate in product and intervention development.


*Trialling work we have been doing with people with dementia, with the product before it actually hits the shelf…we have challenged a couple of the technology companies, you know you are selling this, to people, to individuals but there have never been any focus groups done (WS1P03)*

*It’s that barrier, the people that are designing the technology are not sitting in our world (WS1P05)*


Participation is both a right of itself and a central principle in supporting people’s human rights. The potential of technology to support people to participate in social life and to stay part of their families and communities even when stretched across continents has been demonstrated. However, the participation of older people in the design, development and implementation of technology was less evident, prompting the inclusion of items in the T&SCon guidance which highlight the importance of paying specific attention to this more neglected aspect of participation.

### 3.2. Accountability

Accountability is the second of the five PANEL principles. If we are to argue that technology is increasingly vital in supporting older people’s rights to a family life and to protection from discrimination then accountability, in terms of monitoring impacts of technological development on people’s human rights and providing means of ameliorating any negative impacts on those rights lies with a range of key actors. These include state-level actors who have the power to monitor and where necessary regulate the behaviours of technology companies and promote policies which address structural inequalities and organisational actors who supply equipment and devices and who arguably have a duty to support human rights through self-monitoring and the provision of equitable access to services and to devices which meet users’ accessibility requirements.

Findings from the T&SCon project suggest that devices may be accessible for older people but that often the guidance and help provided in connection with them falls short of what is needed:


*My memory is bad and what I want is something for the iPad that I can just open and read at my own pace…they say you can find everything out from the iPad but I am not au fait with the iPad…I keep forgetting (WS01OP02)*


And that despite policy commitments, broadband access was not universal across Scotland:


*The people who are geographically the most isolated are potentially the people where the infrastructure to support the technology connections is also lagging behind some of the more built-up areas (WS01P03)*


This notion of accountability also extends to organisations ensuring that their websites are developed with accessibility in mind, so that people can easily access their services or navigate to the information they are looking for. This requires website developers to consider accessibility in terms of impairments (e.g., sight loss), computer literacy, the range of devices on which web-based information might be accessed by users, and issues with data cost and internet speed which might also differentially affect older people.

In conclusion, there is work to be done to ensure the accountability of key players in Scotland and elsewhere and to ensure that access to technology is more equitable. There is ongoing debate around whether the access to the Internet should be considered a basic human right. During the COVID-19 pandemic restrictions imposed on people’s movement and engagement with others have meant that technology has often been the only meaningful way to access family and cultural life. In this situation the right to access to the Internet would align with the fundamental rights afforded to everyone with implications for the state and organisational actors accountable for maintaining human rights in relation to access to internet-based services.

### 3.3. Non-Discrimination and Equality

The third PANEL principle relates to contribution of non-discrimination and equality to upholding human rights. The T&SCon project findings illuminated individual differences between people in regard to using technology and revealed how wider stereotypes may lead to assumptions about older people’s engagement with technology and subsequently to discrimination and inequality of access. Equality of access to technology is also impacted by the infrastructure in place to deliver internet access and by individual resources.


*If we don’t have an underpinning understanding of how that impacts on a person’s rights we can all run in different directions with these great new apps, or whatever it is, and the people get lost within that (FG1P02)*


Some older participants actively engaged with technology and recognised its relevance to their lives while others were more reluctant to use technology. Participants across the focus groups ranged in age from their 20s to their 80s and it was clear that age was not the main factor in whether or not people engaged with technology—there were many individual differences demonstrated across age groups. An activity undertaken in the first co-production workshop that involved reviewing terms associated with technology suggested that participants in most age groups were broadly cognisant of everyday technology, including tablets and smart phones. An older person in her 80s was motivated to stay up to date:


*Technology wasn’t in our time, but it is in our time now…when I retired I realised I was going to be left behind, I had stayed with everything during my working life but realised that I would be left behind, so I went out and bought all the gear (FG01OP01)*


Workshop participants identified some age-related conditions that may impact on the ability of people to engage with mainstream technology including dementia and ageing-related sensory loss.


*Devices not just for a person with dementia, they might be living with dementia and living with hearing loss, lots of condition (WS01P03)*


Somewhat conversely, the older person with sight loss who took part in the project was using more complex technology due to their sight loss. This included a table microphone that he brought to the workshop to ensure that he could hear the discussions clearly. His visual impairment had encouraged engagement with technology. Similarly, the range of positive examples shared by a stakeholder from a dementia organisation demonstrated the positive ways that technology can be used by people with dementia.

For organisations promoting and support the use of technology for older people it was important that a person-centred approach was taken to ensure the technology was meaningful and useful for the user. When asked to rank a series of statements about what motivates people to use technology the need for technology to be practical and meaningful was ranked highest.


*Making sure that when people get on the technology that there is something meaningful there, they are not just confronted with a load of apps and thinking, what the? If I press that am I going to break it? (FG02P01)*


These findings illustrate the individual nature of engagement with technology and need for a personalised approach.

Participants also stressed that older people may be limited in the resources they can invest in technological devices and in affording associated costs such as mobile data costs or costs of internet service provision.


*A lot of adults in particular are not able to afford certain types of technology, especially people who are in poverty and maybe they are even more isolated (FG02P02)*

*We were contemplating supplying [tech] to people and did a few questionnaires, asking who had broadband and a high number of people didn’t, and it can come down to the cost (FG1P05)*


More specialist equipment can be more expensive, and continual development of devices can mean that equipment is relatively quickly less or unable to access recently produced content or data formats, incapable of being fully upgraded, or no longer supported by manufacturers.


*[Technology] keeps going up a mark, mark one, mark two, the technology changes (WS1OP01)*


The T&SCon data highlighted significant variability in the quality of the internet signal in different parts of Scotland, illustrating how infrastructure affects use of technology.


*If the broadband isn’t there, then everything fails (FG1P01)*



*See on the west side, you can’t even get a mobile signal let alone… (FG2P03)*



*The people who are geographically the most isolated are potentially the people where the infrastructure to support the technology connections is also lagging behind some of the more built up areas (FG04P01)*


Where health and social work services plan to use technology in the provision of care and support it is important that the organisations invest in the technology and have developed a plan in partnership with the health and care teams for how and where it should be utilised. Only some health board areas in Scotland had access to Attend Anywhere/Near Me video consultation when the T&SCon project took place in 2018. Patients in rural areas without access to this service who needed to access treatment in specialist clinics located in urban areas had to travel, in some cases several hours each way to attend appointments, while clinics local to the specialist clinic could often connect with them online. It is probable that responses to the COVID-19 pandemic have prompted significant changes to this situation, and we reflect on this further in the discussion.

These T&SCon project findings demonstrated significant inequality of access to technology and to the wider infrastructure and systems such as the internet that support the use of that technology and suggested that work was still to be done to ensure equity of access and to address the digital divide in Scotland.

The PANEL principle of non-discrimination and equality underpins recommendations in the T&SCon guidance to focus on people as individuals and to take into account systems and infrastructure. Focusing on personalised use of technology will help to ensure non-discriminatory approaches, improve equality in technology provision, and lead to more inclusive technology-enabled options for staying socially connected.

### 3.4. Empowerment

In the context of older people’s technology use empowerment, the fourth PANEL principle, is met through processes which provide older people with the knowledge, opportunities and skills needed to understand and balance the risks and benefits and make their own decisions about engaging with technology.

Risk is often cited as a barrier to technology use, but when asked to rank potential barriers T&SCon project participants rated risks associated with being connected online as low. Participants were aware of potential risk but this was not perceived to be a significant barrier to using technology. While many risks were identified, there was also a reminder from participants to not over-emphasise risks and a belief that, with training, technology could be used safely by most people. The risk of accessing distressing content and likelihood of financial scams or trolling were seen as low, with none of the older people reporting any experiences of this.


*A lot of perception of the technology is the risk (FG1P01)*



*Need to be aware of this…but not overly concerned (FG03OP04)*


There was also evidence that the older people taking part in the project already had awareness of specific risks and knowledge of how to stay safe online.


*Sometimes you get something by email and you know it is a hack or a scam and you know you need to delete it immediately (FG03OP06)*


There was an indication that using publicly-provided health systems may be perceived as safer as a venue for discussion of sensitive topics and they are more secure than more widely available options.

The T&SCon findings suggested that by equipping them with an understanding of, and the skills needed to address risk older people could be empowered to make their own choices about using technology and specifically around connecting with others online through different platforms. To ensure that this aspect of a rights-based approach is met and to support older people to make their own decisions it is crucial to provide meaningful and accessible training about the use of technology. Project participants recognised that perceptions of risk differ between different groups and that there is a need to both ensure and respect that the older person is making their own choice about risk. The older people taking part in the project demonstrated how training had supported them to have the knowledge they need to assess risk and recognise when they may be at risk online.

Examples were given of both good and poor approaches to training as well as ideas on the best way to present information. Professionals reported that it was important to visit and talk to a person in order to build relationships and to develop trust before introducing new technology. They suggested building up use of technology over time and stressed the importance of ensuring that technology use is meaningful for the user.


*When introducing technology start with something people can relate to, such as old pictures, once you start to show them things, it starts a conversation (FG02P02)*


Older people reported that general technology support such as product helplines and general guides were less helpful. People manning helplines may not be trained to support people with additional needs or those with a lack of experience with technology.


*Computer people talk away in their lingo…whereas if you are doing it you find out the right way…let them use it and keep them right (WS1OP01)*

*You buy a dummies guides…and you get past chapter one and suddenly you’re into the deeps of the depths of it and you’re lost…what we want is something to get us started with the things we want to do (FG03OP05)*


There were many examples of guidance that is too complex or incorrect assumptions that devices can be used ‘intuitively’, leading to a clear recognition that many older people need training and support to use technology. Simple written instructions were perceived as useful by older T&SCon participants as these can be looked at repeatedly. Some of the older people experienced short term memory problems that meant they often needed to refer back to instructions. Practical demonstrations were also regarded as helpful, especially if someone is available to answer questions.


*Demonstrations and hands-on training worked best with some follow-up with simple instructions (WS2OP01)*


For many participants, family members were seen as important in supporting older people to use technology. This often involved intergenerational support with older people describing the help they get from their grandkids, however that was not always the case as demonstrated in the quotes below.


*I had to teach my daughter at 65 to use an iPad (FG1OP01)*

*Two of my sons work in computers so I just give one of them a call (FG3OP04)*

*I have a grandson who helps me (FG3OP01)*


Many older project participants had experience of delivering peer to peer training in technology, and this was seen as a particularly effective way of providing training about technology. While for some, as above, grandchildren could be a great support, for others there was a preference to learn from someone in the same generation.


*Younger generation…it is so second nature for them it can be difficult for them to explain to others (WS1P05)*


The findings from the project demonstrate the importance of providing appropriate training and information to encourage and support the use of technology, empowering older people to make their own decisions about how and when to engage with technology. In response to these findings the T&SCon guidance provides recommendations about training as a key part of enabling choice in relation to the use of technology.

### 3.5. Legality

Participants in the T&SCon project did not engage directly or explicitly with the concept of legality, the last of the PANEL principles, in relation to the provision of technology but there was discussion of risk, as noted above, that included concerns about protecting information about individuals online. Article 8 of the Human Rights Act 1998 stresses the right to privacy in our family lives and the safeguarding of personal information. As such, the legal aspects of handling data held online is central to ensuring a rights-based approach to the use of technology. There are legal frameworks in Scotland that provide clear guidance on how to do this and this aspect of providing technology is considered in the T&SCon guidance section on risk.

This section may be divided by subheadings. It should provide a concise and precise description of the experimental results, their interpretation, as well as the experimental conclusions that can be drawn.

## 4. Discussion

Responses to the COVID-19 pandemic within Scotland and beyond have accelerated the use of technology to support social connectedness for older people within families and organisations [38,39,40]. These rapid changes have taken place in response to an immediate need for families to remain connected and for health and social care services to continue to provide treatment, care, and support. The impact of the pandemic is likely to continue for some time to come. Many of the changes in the way people connect with others that started in response to the pandemic are likely to continue and this is an appropriate time to reflect on the way in which this has taken place and to argue for and support a rights-based approach to how technology is used by both individuals and organisations going forward.

This section first looks at the recommendations that emerge from this PANEL-informed analysis and which frame the pre-pandemic T&SCon guidance. Then we consider how the guidance fits within the wider context of relevant policy and practice in Scotland during the pandemic and moving forward as well as the broader lessons that can be drawn for organisations around the world supporting older people to stay connected by using technology.

### 4.1. Recommendations Emerging from the PANEL Analysis

The use of PANEL principles to guide the analysis of our data enabled us to identify if and how pre-pandemic practice in Scotland promotes a rights-based approach to using technology to support older people to stay socially connected. Using these data alongside findings from the evidence review and secondary data analysis (reported in full elsewhere [1]) we derive a set of recommendations that underpin the T&SCon guidance and that have relevance beyond Scotland.

Focusing on participation and non-discrimination highlights the importance of seeing older people as individuals and of challenging stigma and stereotypical views of older people when designing technology interventions. Ageism has been exacerbated during the pandemic [41,42] further emphasising the need to ensure older **people** are seen as individuals. Our participants included older people who were enthusiastic about using technology and very able to engage with a range of devices but others who were less confident and required specific support and/or adaptations to the technology.

This links to the need for appropriate **training** and support for older people when being introduced to a new piece of technology. Older people in this study identified limitations to current helplines and instructions provided for the general public, underscoring the need for tailored and accessible training and guidance documents. Peer support was identified both by T&SCon participants and in the literature [43] as an effective way to support older people in the use of technology.

Supporting older people’s **participation** in the development of technology interventions will further ensure that older people are recognised as individuals, ageism is challenged and appropriate training and support are provided. Co-production approaches are now commonplace in service development and research [44,45] but our data here suggest that there is still scope to significantly improve the inclusion of older people in the design of technological interventions to support social connectedness. Examples from this project highlight the important role that older people can and do play in the support and training of other older people to use technology.

The older people and professionals taking part in this study all recognised the **risks** associated with activities undertaken online relating to issues such as financial scams, trolling and theft of personal data. However, they, particularly the older people, stressed the importance of clear and balanced information about these risks to avoid an overly risk-averse approach that buys into ageism and stereotypical ideas about all older people as vulnerable. Older people have a human right to make decisions about their own lives and these include decisions about the use of technology.

To enable any of this to happen older people need to have access to the necessary technological devices and **systems**. Participants in this research noted that older people might be excluded from the use of technology, for example due to the cost of devices and Internet access and due to geographical location [9]. This issue is discussed in more detail in the section that follows.

The PANEL analysis presented in this paper provides additional support for the recommendations underpinning the T&SCon guidance which are presented in Figure 5 below.

These recommendations frame the T&SCon guidance and provide a practical approach to ensuring a rights-based approach is taken when supporting older people to use technology to stay connected. Thus, we would argue that despite primary data collection taking place before the pandemic our guidance provides a broad framework for organisations working in this peri-COVID world to ensure the human rights of older people are upheld at a time when they are threatened through structural and cultural responses to COVID-19.

### 4.2. Supporting Older People to Stay Connected during the Pandemic in Scotland

The T&SCon guidance contributes to wider policy and practice initiatives in Scotland that aim to support older people to stay socially connected and that have evolved rapidly in response to the COVID-19 pandemic. Recent initiatives reflect pre-pandemic policy that recognised the risks of social isolation and loneliness for older people [18], put an emphasis on protecting human rights [31] and supported the use of technology [46].

Since the first pandemic responses in March 2020 there has been high-level recognition of the challenges and social isolation experienced by citizens alongside organisational responses to address digital poverty and digital exclusion. These include policy initiatives, investment of funding and reorganisation of health and social care services. Key policy and strategy documents in Scotland which have dealt with the impact of the COVID-19 pandemic on our connectedness as citizens include *Protecting Scotland, Renewing Scotland* [47] and the Digital Scotland strategy *A Changing Nation: How Scotland will Thrive in a Digital World* [40]. A common thread running through the priorities outlined in these policies is around digital inclusion, citizen participation and supporting people to use technology effectively to access services.

Ofcom [48] state that only 1.2% of households in Scotland are without a ‘decent broadband connection’; however, these houses are often in remote areas where the cost of installing a connection may be high. This potentially results in isolating people who may be most in need of an Internet connection to stay socially connected. Since the early 2000s, there has been continued debate on whether to consider access to the Internet as a basic human right. In 2016, the United Nations General Assembly passed a non-binding Resolution that declared internet access a human right [49] but, as a non-binding resolution, this does not compel governments to act. In the UK in March 2020 Ofcom established the legal right for ‘every home and business in the UK to request a decent, affordable broadband connection’ [50]. It is hoped that this right will ensure universal coverage in coming years, however, ongoing costs for devices and data will need to be met by individuals and these costs will continue to exclude people and perpetuate digital divides.

A key practical element in the Scottish response has been the Scottish Government’s *Connecting Scotland* initiative, a partnership between the Scottish Government, the Scottish Council for Voluntary Organisations, industry, and Local Authorities. The objective of this scheme was to provide those currently without access to the internet with equipment and support to access services and connect with friends and families online. The objective of this programme is to address digital exclusion, which significantly disadvantages those who could make use of digital tools to access services and to support social connection. Work has also been undertaken to support the care home sector, a sector very significantly impacted by COVID-19 [51].

There is recognition that in Scotland accelerating digital adoption in response to the pandemic has resulted in many benefits including the shift to remote working for many staff in the health and social care settings. In service provision, the rapid rollout of video consultation and other new ways of working, have all provided access to services and maintained provision in challenging circumstances. Health and social care services play a key role in wellbeing for older people and in their social connections and there is ongoing concern in Scotland that people’s human rights have the potential to be affected by increased use of technology in care provision contexts. Respondents to a recent citizen’s panel report had variable experiences of using telephone and online consultation during the pandemic but the majority indicated that they would be willing to see a health or social care professional via video consultations (64%) and via telephone consultations (58%) if it meant health services suspended during the pandemic could resume [52]. However, those in the 65+ age group were less likely to say they would use video or telephone consultations or an app, text, or website. This underlines the need for guidelines such as those produced in the T&SCon project to improve the way technology is used to connect older people to others and is further addressed through a collaboration of third sector organisations who have developed a set of guiding principles: *Human Rights Principles in Digital Health and Social Care* [53]. The authors of this report share the concerns raised within the T&SCon project about digital exclusion, inequity in provision, privacy, and costs.

In May 2021 the Scottish Government’s Digital Citizen Delivery Plan [54] was published. The plan recognises that barriers to connectedness remain and that the accelerated adoption of technology has also amplified digital exclusion. Indeed, as we move through the phases of responding to COVID-19 from acute response to longer term planning, there is greater interest in the potential of using technology to facilitate access and support remobilisation of services and individual resilience.

The Strategic Priorities identified within this document for 2021-22 are:▪Addressing Inequalities and Promoting Inclusion;▪Engaging citizens, staff and services through Co-design and Participation;▪Redesigning Services—Improving Citizen Access/Promoting Wellbeing;▪Innovating to Support Transformation.

These policies and practices will enable the future promotion and protection of older people’s human rights within initiatives promoting the use of technology and alongside these the T&SCon guidance and recommendations provide a practical toolkit for taking these principles and goals forward in practice.

## 5. Conclusions

It is possible to follow a common thread through the way in which state level responses to COVID-19 have shone a light on the lived experience and the quality of life of older people globally. The human rights implications of how technologies may be utilized to facilitate social connectedness and the use of technologies to provide a connection to required health and care services have become a more obvious parts of the discussion.

Our findings, and the reflections on Scottish policy and practice during the pandemic, presented in this paper identified specific challenges in supporting older people to stay socially connected in a world where people are more commonly living alone and where social disconnection has been exacerbated by responses to the COVID-19 pandemic. The risks posed by digital poverty identified in Scotland are likely to be more pronounced in other regions of the world, particularly the global south [55].

Our findings demonstrate the willingness of older people to use technology but emphasize the need for processes to be participatory and responsive to individual needs, taking a rights-based approach to support engagement. Further, to support rights, this process should be about connecting one human being with another human being. The experiences and risks of loneliness and social isolation for older people that were evident before the pandemic and exacerbated by lockdowns and restrictions during the pandemic will continue to need and action attention from policy and practice.

Now that the initial rush precipitated by the pandemic to get online has passed, further reflection is needed to sustain meaningful social connection for older people and the role that technology might place to support that.

## Figures and Tables

**Figure 1 ijerph-18-08725-f001:**
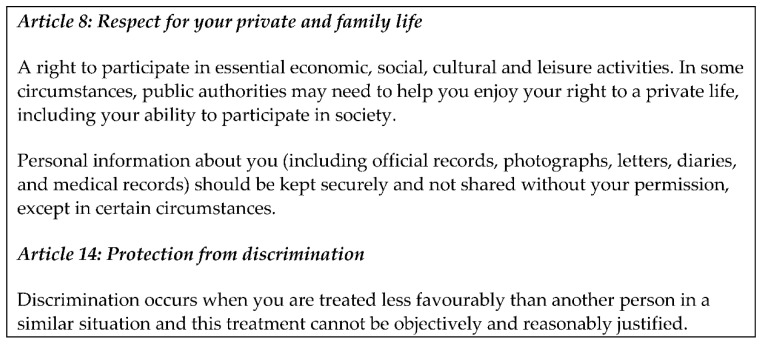
Relevant Articles from the UK’s Human Rights Act 1998.

**Figure 2 ijerph-18-08725-f002:**
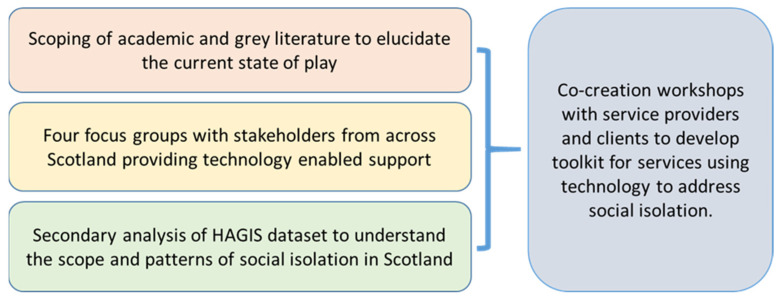
T&SCon Project Methods.

**Figure 3 ijerph-18-08725-f003:**
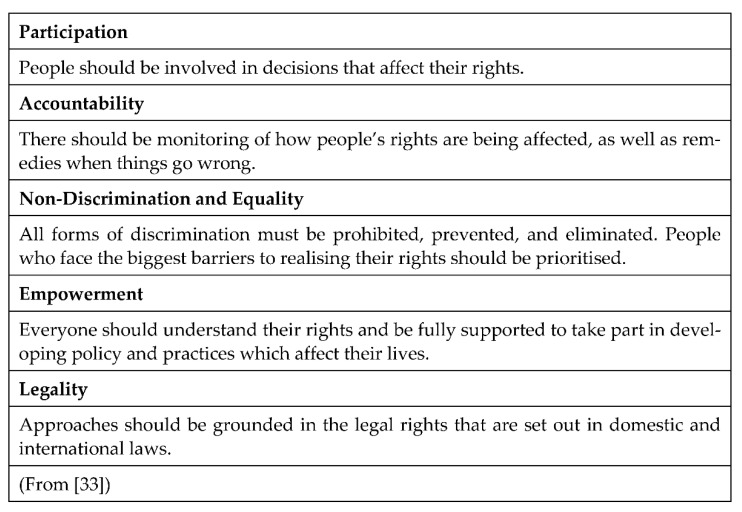
PANEL Approach to Human Rights.

**Figure 4 ijerph-18-08725-f004:**
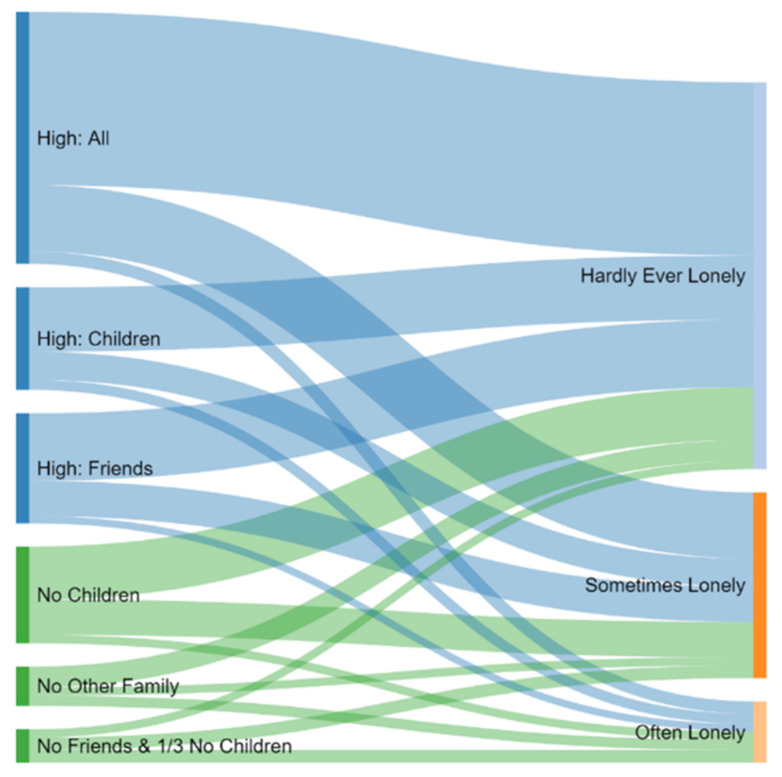
Relationship between social connectedness and loneliness. Source: HAGIS.

**Figure 5 ijerph-18-08725-f005:**
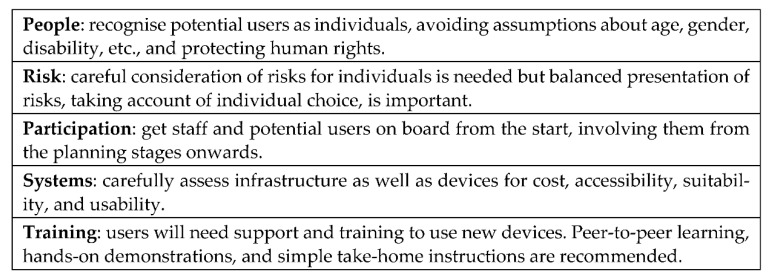
Recommendations from the T&SCon project.

## Data Availability

A partial dataset from this project can be accessed at https://tec.scot/news/2020/05/26/technology-and-social-connectedness/ (accessed on 17 August 2021).

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
