# Peer review of "Using Technology the Right Way to Support Social Connectedness for Older People in the Era of COVID-19"

_ijerph, 2021, doi:10.3390/ijerph18168725_

Round 1

Reviewer 1 Report

This article draft provides some insights on technology use by older adults with some focus on social online connections (but also health-related online activities). Most input is from older adults themselves, with and without impairments, with also some input from caregivers. There is a also secondary data analysis on feeling lonely and social connections.

The topic of digital technology use to support social connections in older age has been relevant before the Covid-19 pandemic and is now in this pandemic relevant more than ever. Thus, this paper adresses an important research question and as main insight shows that older adults need to be included more as participants in the design of technologies as well as in how they are disseminated and taught to the older population. But, as presented, the methodology is too unclear and the exaggerated framing as this study adressing technology use during the Covid19 pandemic is misleading (it was a pre-pandemic study!) so this must be improved in a revision before publication can be considered.

Major issues

  1. This study was conducted before the pandemic. It's results may be relevant to technology use and offers to/by older adults during the pandemic but this can only be a specuative section on possible implications in th discussion. The framing of this study as examining technology support during the Covid-19 pandemic is simply false and as such misleading. Further, when discussing these possible implications it must be carefully considered that use of and attitudes toward technology use in older adults likely have critically changed from the pre-pandemic assessment conducted here.
  2. The sample is not sufficiently described. Results are mostly reported from the older audults. How old were they, what was their educational background, where did they live (it is sometimes alluded to rural areas but it is not clear whether the sample was exclusively composed of people living in rural areas or if not how this was distributed between rural and urban), did they live on their own etc.
  3. The Results section on "Participation" primarily contains general informatio about technology use and reasons for technology use among the participating older adults. The information on "Participation" in the sense of the applied PANEL category is only contained in the last few lines and shows a deficit thereof. The first part should thus be removed to a general overview section to not mislead as if there was a lot on participation in the PANEL sense.
  4. The secondary data analysis from HAGIS and Figure 4 seems quite interesting and informative but I could not fully understand what is displayed from the Figure, there is no appropriate Figure note and the information in the text is also not sufficiet. Curcially, what is the relation to technology use? Is "social connectedness" more specifically defined as digital social connections? If it is, this is informative. If it is not, I do not see how this contributes to the current article focus on using technology to support social connectedness.

Reviewer 2 Report

Interesting paper on the importance of digital inclusion for the elderly and its difficulties, especially in this era of physical distancing due to the COVID-19 pandemic. I consider the text relevant and appropriate for publication. 

My main suggestions for improving the text are:

  1. the manuscript is long, and although the reading is pleasant, summarizing the text would be very interesting;
  2. In the Discussion section, it would be interesting for the authors to address aspects of the "digital inclusion of the elderly problem" in different regions of the world;
  3. In the Conclusion section, it would be didactic to have an objective listing with author's suggestions for improve social connectedness for older people;
  4. In the attached pdf I introduce some other observations that I consider relevant.

Round 2

Reviewer 1 Report

The authors made sensible changes to the manuscript, improving it. I think it is as well presented as possible but the conclusions that can be drawn from this rather qualitative and exploratory pre-pandemic research for the current Covid-19 pandemic and beyond remains limited (hence I would prefer that the Covid-19 era reference is removed from the title). I do think that the contained quantitative analysis is quite interesting. Ultimately, it is up to the Editor to decide whether this contribution is sufficient for this journal. The presentation is good, I don't see need for revision there (I do wonder what it means that three older adults lived in urban areas and "most others" in rural areas - what about the rest then? Isn't it rather almost all of the sample lived in rural areas with three exceptions?)